# Feature Merged Network for Oil Spill Detection Using SAR Images

Yonglei Fan [1], Xiaoping Rui [2,*], Guangyuan Zhang [1], Tian Yu [3,4], Xijie Xu [3] and Stefan Poslad [1]

1   School of Electronic Engineering, Queen Mary, University of London, London E1 4NS, UK;
    yonglei.fan@qmul.ac.uk (Y.F.); guangyaun.zhang@qmul.ac.uk (G.Z.); stefan.poslad@qmul.ac.uk (S.P.)
2   School of Earth Sciences and Engineering, Hohai University, Nanjing 211100, China
3   College of Resources and Environment, University of Chinese Academic of Sciences, Beijing 100049, China;
    yutian@mails.ucas.ac.cn (T.Y.); xuxijie18@mails.ucas.ac.cn (X.X.)
4   Research Institute of Solid Waste Management, Chinese Research Academy of Environmental Sciences,
    Beijing 100012, China
*   Correspondence: ruixp@hhu.edu.cn; Tel.: +86-13671106220

**Abstract:** The frequency of marine oil spills has increased in recent years. The growing exploitation of marine oil and continuous increase in marine crude oil transportation has caused tremendous damage to the marine ecological environment. Using synthetic aperture radar (SAR) images to monitor marine oil spills can help control the spread of oil spill pollution over time and reduce the economic losses and environmental pollution caused by such spills. However, it is a significant challenge to distinguish between oil-spilled areas and oil-spill-like in SAR images. Semantic segmentation models based on deep learning have been used in this field to address this issue. In addition, this study is dedicated to improving the accuracy of the U-Shape Network (UNet) model in identifying oil spill areas and oil-spill-like areas and alleviating the overfitting problem of the model; a feature merge network (FMNet) is proposed for image segmentation. The global features of SAR image, which are high-frequency component in the frequency domain and represents the boundary between categories, are obtained by a threshold segmentation method. This can weaken the impact of spot noise in SAR image. Then high-dimensional features are extracted from the threshold segmentation results using convolution operation. These features are superimposed with to the down sampling and combined with the high-dimensional features of original image. The proposed model obtains more features, which allows the model to make more accurate decisions. The overall accuracy of the proposed method increased by 1.82% and reached 61.90% compared with the UNet. The recognition accuracy of oil spill areas and oil-spill-like areas increased by approximately 3% and reached 56.33%. The method proposed in this paper not only improves the recognition accuracy of the original model, but also alleviates the overfitting problem of the original model and provides a more effective monitoring method for marine oil spill monitoring. More importantly, the proposed method provides a design principle that opens up new development ideas for the optimization of other deep learning network models.

**Keywords:** SAR; oil spill; image segmentation; deep learning; UNet; FMNet

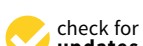

## 1. Introduction

Oil spills in the ocean have become a serious environmental problem causing, long-lasting financial costs and threats to marine life [1]. It was reported that more than a billion dollars were spent and more than 200,000 marine lives were lost during the 2010 Deepwater Horizon (DWH) oil spill [2]. The continuous increase of pollutants from marine oil spills will have an increasingly negative impact on the environment, biodiversity will be reduced, and eventually the ecosystem imbalance will endanger human survival and sustainable development. Oil spill monitoring can quickly and accurately determine the occurrence of oil spills and guide the emergency treatment of environmental pollution to minimize the

harm caused by such spills. With the frequent occurrence of oil spill incidents, accurate, efficient, and automatic marine oil spill monitoring has become a necessity.

Synthetic aperture radar (SAR) image-based oil spill detection is the most commonly used and effective monitoring method [3]. As an active side-looking radar system [4], the imaging geometry is an oblique projection type. The working wavelength, incidence angle, polarization mode of the radar sensor, surface roughness, and dielectric constant of the ground object [5] all affect the backscattering of the signal. The polarized electric field vectors in the horizontal and vertical directions form different polarization phenomena [6]. The backscattering intensity of a rough surface is higher than that of a smooth surface. The surface of the oil spill area with no obvious wave effect is much smoother than that of sea water, and thus the oil spill area is shown as black pixels in the SAR image, whereas the sea water surface is shown as bright pixels. Therefore, a low regional pixel value indicates that the backward reflectivity of this part is low, and the probability that this region is an oil spill area or an oil spill like area is high. The ultimate purpose of the model is to better distinguish oil spill areas from oil spill-like areas. The model not only needs to use pixel features to classify the overall light and dark areas, but also needs to obtain the shape, boundary and other features of each category in order to complete the segmentation results with high accuracy. However, a large amount of speckle noise negatively interferes with SAR image target detection, which significantly reduces the accuracy of image segmentation or edge extraction, posing a great challenge to the effective monitoring of marine oil spills. In addition, the areas with a similar appearance as oil spills, like the red part of Figure 1b, also have a black spot effect on SAR images, which makes it more difficult to ascertain such areas correctly.

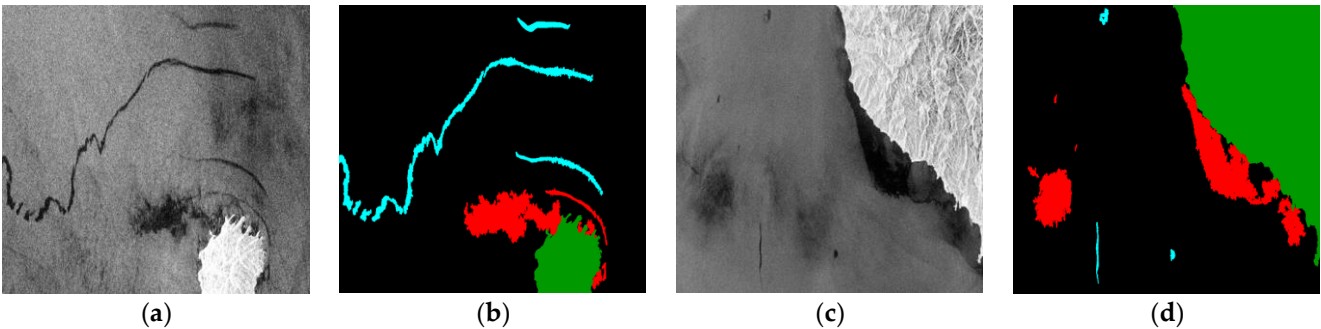

| (a) | (b) | (c) | (d) |

**Figure 1.** Two examples of source data, (**a**,**c**) are SAR images and (**b**,**d**) are 'image-segmentation' labels. Cyan corresponds to oil spills, red to look-alikes, brown to ships, green to land, and black is the sea surface.

There are two main ways to identify an oil spill area i.e., manual extraction and automatic extraction. With the rapid development of deep learning theory and practice and its unique and powerful image recognition capabilities, most research teams in recent years have focused on the use of deep learning methods to automatically identify oil spill areas in SAR images. In 2000, Del Frate et al. [7] proposed the use of neural networks to identify oil spill areas. At that time, only simple weight connections were used to build the network. Although the effect was not very good, the idea of automation was enlightening. Fiscella [8] developed and tested an automatic classification algorithm based on probability statistics to obtain an oil spill area. Targeted feature extraction is conducted on dark spots in SAR images and can replace manual inspections for large ocean areas. A fast SAR image segmentation method based on the artificial bee colony algorithm [9] was proposed to solve the existence of speckle noise in SAR images. In [10–12], the authors used the spatial density function, a kernel Fuzzy C-means clustering (FCM) algorithm, and the spectral clustering (SC) ensemble algorithm to extract dark area features, respectively. For feature extraction in dark areas, SAR images can be understood intuitively. However, with the development of machine learning, Topouzelis [13] proposed an oil spill feature selection and classification technology based on decision tree forests. In addition, Yu et al. [14] proposed the adversarial

learning of an f-divergence function in 2018. In the same year, Orfanidis [15] proposed the use of a DCNN for SAR image segmentation. In 2019, Gallego [16] and Krestenitis [17] proposed an end-to-end deep learning image segmentation algorithm. In summary, the segmentation of the oil spill area in SAR images is mainly divided into specific feature analysis and extraction and deep learning-based complex and uncertain feature extraction. As a disadvantage of a segmentation method based on specific characteristics, oil spill areas, oil-spill-like areas, and other types of ground features cannot be completely distinguished through only a few features. This is the main reason for the low accuracy of a segmentation method. Based on the deep learning algorithm for SAR image segmentation, the model can extract and combine high-dimensional features to classify different ground feature types, which greatly improves the accuracy of the segmentation.

In this study, the FMNet semantic segmentation model is proposed to improve the accuracy of marine oil spill area monitoring. Firstly, a threshold segmentation algorithm is used to process the original data to obtain the high frequency the information of the original image in the frequency domain. This part of information represents the boundary between the categories within the image. This traditional image processing method obtains the approximate global characteristics of the image, and at the same time weakens the influence of intra-class noise. Then the convolutional neural network is used to extract high-dimensional features from the global features, and the high-dimensional features of the original image are combined for crossover, complementing the advantages, and finally providing better decision-making for the segmentation model. Five commonly used threshold segmentation methods are compared in this study, and the adaptability of each threshold method to high-frequency SAR image feature extraction was explored. The extracted high-frequency feature information and SAR image source data were input into the deep learning semantic segmentation network, and the network structure was debugged to adapt the model to the data.

## 2. Materials and Methods

In this section, the source, processing, and numerical characteristics of the dataset are introduced. The design idea and theoretical basis of the model, including a description of five commonly used image threshold segmentation methods and characteristics, followed by two different deep learning model frameworks, are enunciated then validated.

### 2.1. Dataset

There has long been a lack of public or industry recognized standard oil spill datasets. Although algorithms from different research teams [18–21] can be used to compile better image segmentation results on their respective private datasets, large errors occur when well-directed methods are applied to other studies. Therefore, a standard industry dataset is needed to ensure that the research results of each research team have a unified measurement standard.

The oil spill detection dataset from the European Space Agency (ESA) was created and provided to the scientific community by Krestenitis et al. in their work [17,22] and can get from the website [23]. It has been used by a large number of research teams in recent years. The oil spill detection dataset contains jpg images extracted from satellite SAR data depicting oil spills and other relevant instances, as well as their corresponding ground truth masks. The initial SAR data were collected from the ESA database, the Copernicus Open Access Hub, acquired through the Sentinel-1 European Satellite missions. The required geographic coordinates and time of the confirmed oil spills were provided by the European Maritime Safety Agency (EMSA) based on the CleanSeaNet service records, covering a period from 28 September 2015 to 31 October 2017.

Each oil spill area in the SAR images of the dataset was accurately located and recorded by the ESA. All images were then re-scaled to achieve a pixel resolution of $1250 \times 650$. A speckle filter was used to reduce the sensor noise, and a $7 \times 7$ average filter was used to suppress the salt and pepper noise. The developed dataset (~400 MB) contains

approximately 1002 images for training and 110 images for testing, depicting instances of five classes, namely oil spills, look-alikes, land, ships, and sea areas. The main purpose of the division of the five classes is to reduce the impact of land and ship regions on the classification.

The difficulty of oil spill detection is to distinguish oil spill areas from oil-spill-like areas. By analyzing the dataset, it can be seen that in some images, a large oil-spill-like area is mixed with a small actual oil spill area, as shown in Figure 1. In addition, in some other images, the characteristics of the oil spill area and oil-spill-like area are almost the same, and thus it is difficult to distinguish them through human recognition, as shown in Figure 2. These difficulties greatly increase the aporia of the model recognition, which is the main problem faced by research teams.

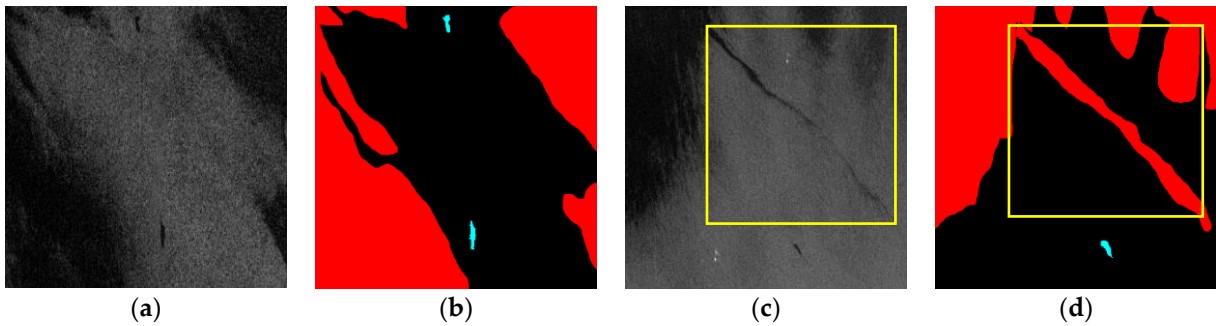

| (a) | (b) | (c) | (d) |

**Figure 2.** (**a**,**c**) are samples of SAR images with large oil-spill-like areas and small oil area inclusion, (**b**,**d**) are labels. The long stripe area (in the yellow square) may cause some problems in the model due to the relatively asymmetric mark.

### 2.2. Algorithm Theory Analysis and Design

The disparity between the accuracy of the training results and the test results of the existing models indicates that the overfitting problem of the algorithm might be conspicuous. From this point of view, an innovative model based on the analysis of theoretical knowledge was designed in this study to reduce the overfitting problem of the model. Insufficient datasets, an inconsistent feature distribution, and excessive sample noise may lead to an overfitting of the final results of the algorithm. SAR images show the characteristics of speckle noise in areas without oil spills. This type of noise contains a large number of pixels with the same backscattering intensity as the oil spill area, which significantly interferes with the image feature extraction of the network model during the convolution process. In this study, the characteristics of noise in datasets are analyzed, and the traditional image threshold segmentation technology is proposed to process the datasets, classify the datasets from the numerical features, extract the high-frequency information from the datasets, highlight the global features of the image, and serve as the feature input layer of the model. At the same time, the deep learning network model extracts the high-dimensional information features of the data from the source dataset; that is, the detailed features, as the source data layer of the model. By combining two feature input layers to build a deep network model, an innovative deep learning network model structure is formed. The algorithm exhibits an overfitting phenomenon. The reasons include fewer datasets [24], an inconsistent feature distribution [25], and excessive sample noise [26]. This algorithm analysis mainly focuses on the algorithm design and optimization in terms of excessive noise in the dataset. At the front end of the network architecture, the traditional image segmentation method was first used to extract the dataset features.

### 2.3. Global Feature Extraction Methods for SAR Image

The purpose of image thresholding is to divide pixel into different sets according to the gray level, and each subset forms a region corresponding to the real scene. Each region has the same attribute, but adjacent regions do not. Such a division can be achieved by selecting one or more thresholds from the gray level. High-frequency information is expressed, and low-frequency information is weakened during this process. In a simple

form, the image masks the detailed features and highlights the global features. This study compares the effects of five commonly used threshold segmentation techniques on SAR image segmentation.

### 2.3.1. Binary Threshold

Segmentation principle [27]: Select a specific threshold, and set the gray value of the pixel points greater than or equal to the threshold to the maximum value of 255, and the gray value of the pixel points smaller than the threshold value to zero, as shown in Equation (1) and the schematic diagram in Figure 3b.

$$dst(x,y) = \begin{cases} maxVal, & if\ src(x,y) > thresh \\ 0, & otherwise \end{cases} \tag{1}$$

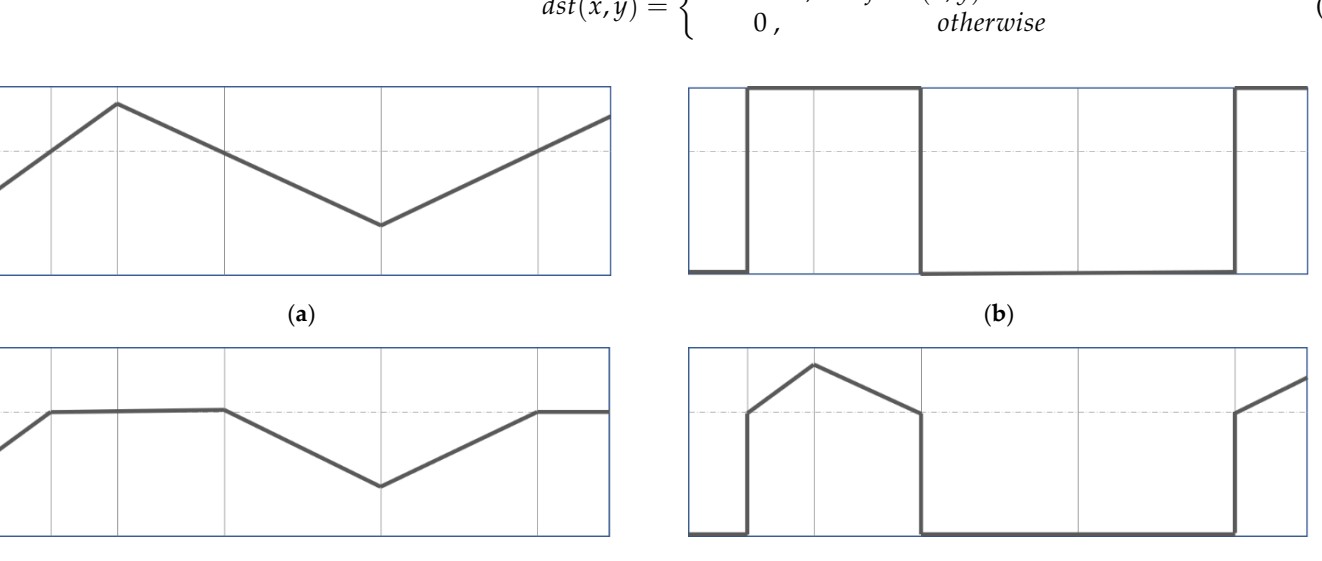

**Figure 3.** (**a**) Assumed pixel value distribution, (**b**) distribution after binary threshold, (**c**) distribution after the truncated threshold, and (**d**) distribution after ToZero threshold.

### 2.3.2. Truncate Threshold

Segmentation principle [28]: A user-defined threshold value first needs to be selected. The gray value of the pixel points in the image greater than or equal to the threshold value is set as the threshold value. The gray value of the pixel points that are less than the threshold value remains unchanged, as shown in Equation (2) and the schematic diagram in Figure 3c.

$$dst(x,y) = \begin{cases} threshold, & if\ src(x,y) > thresh \\ src(x,y), & otherwise \end{cases} \tag{2}$$

### 2.3.3. ToZero Threshold

Segmentafion principle [29]: Select a threshold value customarily, where the gray value of pixels greater than or equal to the threshold value remains unchanged, and the gray value of pixels less than the threshold value is set to zero, as shown in Equation (3) and the schematic diagram in Figure 3d.

$$dst(x,y) = \begin{cases} src(x,y), & if\ src(x,y) > thresh \\ 0, & otherwise \end{cases} \tag{3}$$

### 2.3.4. Triangle Threshold

Segmentation principle [30]: This method uses histogram data to find the best threshold based on the geometry method. Its established condition assumes that the maximum wave peak of the histogram is near the brightest side, and the maximum linear distance

d is then obtained using a triangle. The gray level of the histogram corresponding to the maximum linear distance, b, is the segmentation threshold. A schematic diagram is shown in Figure 4.

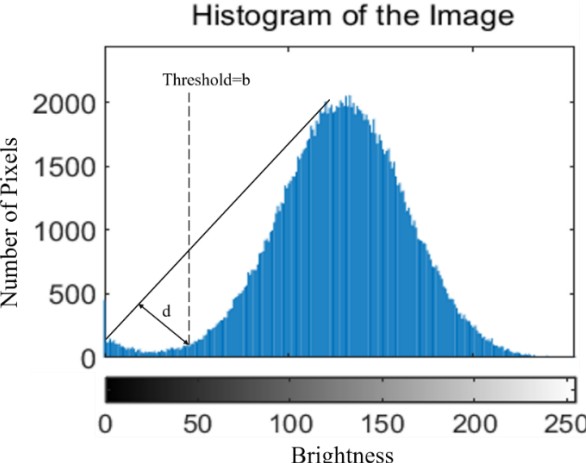

**Figure 4.** Determine the threshold geometrically using the histogram.

### 2.3.5. OTSU Threshold

The maximum inter-class variance, proposed by Otsu in 1979, is an adaptive threshold determination method [31]. The algorithm assumes that the image pixels can be divided into background and target parts according to the threshold. Here, $\omega_0$ represents the proportion of target pixels in the image, $\mu_0$ is the average gray value, $\omega_1$ is the proportion of background pixels in the image, and $\mu_1$ is the average gray value. The average gray value of all pixels in the image is represented by $\mu$, and the variance between classes is represented by $g$. The optimal threshold T is calculated using Equation (4) to distinguish the two types of pixels, which makes the maximum discrimination between these two pixels. This global binarization-based algorithm is simple and fast and is unaffected by image brightness and contrast. The disadvantages of this algorithm are that it is sensitive to image noise and can only segment a single object. When the ratio of the target and background is significantly different, the variance function between classes may exhibit a double- or multi-peak phenomenon, and the segmentation effect will become poor.

$$g = [\omega_0\omega_1(\mu_0 - \mu_1)]^2 \tag{4}$$

where $g$ is the variance between classes, $\omega_0$, $\omega_1$ are the ratio of foreground and background pixels to the entire image, $\mu_0$, $\mu_1$ are the average value of foreground and background pixels.

### 2.4. Introduction of Deep Learning Algorithm

In 2015, Ronneberger et al. [32] proposed a UNet network structure, which greatly promoted research on medical image segmentation. UNet is based on the expansion and modification of a fully convolutional network. Figure 5 demonstrates the framework of UNet [32]. The architecture is characterized by end-to-end image segmentation technology, and in the up sampling process, the depth features obtained by convolution operation are used as an important basis for each up sampling decision. The network consists of two parts: a contracting path to obtain context information and a symmetrical expanding path to determine the position. The entire network has 19 convolution operations, 4 pooling operations, 4 up-sampling operations, and 4 cropping and copying operations. The convolution layer uses the "valid, padding = 0, stride = 1" mode for convolution, and thus the final output image is smaller than the original image. When the convolution operation uses

the "same" mode for processing, the output image with the same size as the original can be obtained. The structure of the model loss function is described in Equations (5) and (6):

$$p_k(x) = \exp(a_k(x)) / \sum_{k'}^{K} \exp(a_{k'}(x)), \tag{5}$$

$$w(x) = w_c(x) + w_0 * \exp\left(\frac{-(d_1(x) + d_2(x))^2}{2\sigma^2}\right), \tag{6}$$

where $a_k(x)$ represents the value of position $x$ in the kth channel, and $K$ represents the number of categories. In addition, $w_c$ is the input segmentation image mask, $d_1(x)$ represents the distance from pixel $x$ to the first category closest to it, and $d_2(x)$ represents the distance from pixel $x$ to the second closest category.

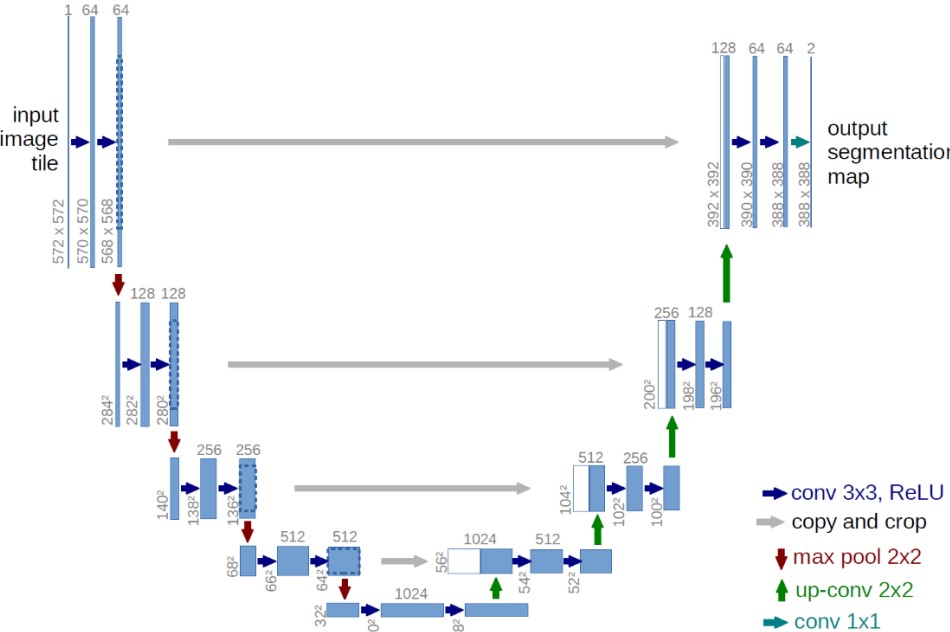

**Figure 5.** U-net architecture [32] (example for 32 × 32 pixels in the lowest resolution). Each blue box corresponds to a multi-channel feature map. The number of channels is denoted on top of the box. The x-y-size is provided at the lower left edge of the box. White boxes represent copied feature maps. The arrows denote the different operations.

Through the processing of this algorithm, the weight of pixels at the boundary of the category will be larger, and the weight of pixels farther from the boundary will be smaller. The reason for this is that, within the category, the pixel features are similar; therefore, this similarity should be weakened. The boundary area between the different categories was the main influencing factor for segmentation. Therefore, it is necessary to strengthen the boundary features and provide greater weight to make it easier after training. Moreover, the segmentation results are more accurate.

### 2.5. Feature Merging Network (FMNet)

According to the design idea, first, the source data are segmented by a threshold. The threshold segmentation divides the levels according to the gray value of the pixel and uses a simple clustering principle to distinguish the different categories from the numerical value. The function is to extract the global features of the image and strengthen the local features. After threshold segmentation, the texture features in the source data image are highlighted, the boundaries between categories are clearer, and the global features of the source data are enhanced. In addition, local features within the category are weakened. Because the pixel

values within the same category are similar, similar pixel values are converted similarly through the threshold, such as in Equation (1), which reduces the impact of noise within the category. In this study, we use the above five threshold segmentation methods to extract the global features of the image and combine them with the depth convolution network to build the FMNet model.

The next step is to input the source and feature data into the encoder network and use a convolution operation to extract high-dimensional features. This part of the convolution operation mainly uses a convolution kernel size of $3 \times 3$. The convolution results were then normalized and activated. To improve the receptive field of the model, the maximum pooling operation is then conducted using a $2 \times 2$ pooling core, where the step size is set to 2. At the same time, the index of the position of the maximum value is recorded in the maximum pool, which is used to apply nonlinear up-sampling in the decoder process. Feature maps contain the high-dimensional features of the source data, and the high-dimensional features of the global features are finally output by the encoder network.

Next, the feature maps, namely, the decoder operation, are sampled. The explicit path consists of several blocks, and different decoder networks have different numbers of blocks. Within each block, the size of the input feature maps is doubled, and the number is halved. The feature maps of the left symmetric compression path were then clipped to the same size as the extended path and normalized. The deconvolution kernel size used in the sampling was 2 pixels $\times$ 2 pixels. Finally, the prediction results of k (number of categories, k = 5) channels with the same size as the original image are input into the softmax layer for the final classification.

The following pseudo-code (see Algorithm 1) was completed according to the design process shown in Figure 6. In this study, five threshold segmentation algorithms and a basic semantic segmentation network model, UNet, are combined to form five feature merge semantic segmentation network models. They are compiled through the Keras framework [33], including training and testing. For 1002 SAR images in the training dataset, they were divided into 50 batches, where each batch has 450 iterations, and the total number of iterations is more than 20,000. As a parameter setting of the network that needs to be explained, the feature merging does not affect the parameter quantity of the semantic segmentation model; however, the basic network model is the factor determining the number of parameters. The overall parameter quantity of the UNet model is 3.6 million.

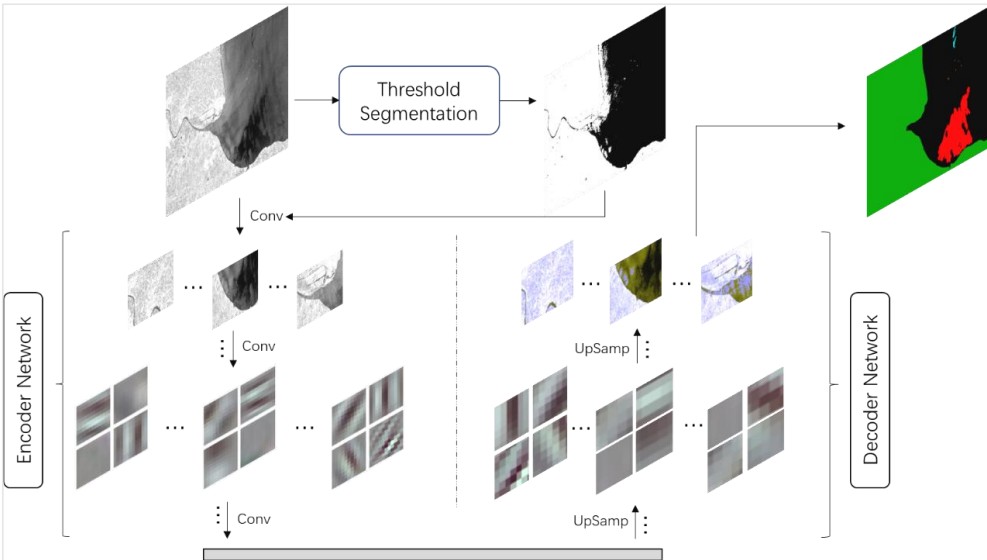

**Figure 6.** Schematic depiction of FMNet architecture used for segmentation. The traditional threshold segmentation algorithm is combined with the deep learning model to form FMnet. The global features provided by the threshold segmentation model reduce the noise for the feature extraction of the depth model, and also provide more clear boundary features in the segmentation process.

The equipment used in this experiment included a GeForce RTX 2060 with a core frequency of 1770 MHz and a video memory capacity of 6 GB. The CPU is a 10,400 version of Intel i5, with a frequency of 2.9 GHz. The memory size of the device was 16 GB.

---

**Algorithm 1.** Pseudo code of the FMNet segmentation algorithm.

---

**Input: Oil Spill Dataset**
  **Function: Threshold Segmentation (TS)**
    If *src(x,y)* > the value of threshold then
      *src(x,y) = 0* or *maxValue* Depending on the algorithm of *TS*
    Return feature image
  **Function: Merge**
    Merge feature image and original image
    Return: fusion data
  **Encoder Network:**
    Repeat:
      Convolutional: $(f * g)(1,1) = \sum_{k=0}^{2} \sum_{h=0}^{2} f(h,k) g(1-h, 1-k)$
      Rectified Linear Unit (ReLU) activation: $a^{(l)} = f\left(W a^{(l-1)} + b\right)$
      Max Pooling: Select the largest value in the kernel and record the position
      Return: feature maps
  **Decoder Network:**
    Repeat:
      Transposed Convolution: Combine the down-sampling features to reproduce the segmented image
      Return: *k* segmentation maps
**Softmax:**
  Decision making: Each pixel position selects the category with the greatest probability
**Output: Segmentation image**

---

Another important network parameter is the learning rate. If the learning rate is too high, the training result may not converge or diverge. The change in the range of weight may be extremely large, making the optimization over the minimum value and worsening the loss function. When the learning rate is too low, the training will become more reliable; however, the optimization process takes a long time because each step toward the minimum value of the loss function is small. This paper compares the initial learning rates of 10-3, 10-4, and 10-5 experimentally, and the results show that the training effect of the 10-4 sized learning rate model is the best. The optimizer chooses the best Adam algorithm and uses cross entropy to isolate the parameters, which also reduces the possibility of an overfitting.

### 3. Results and Discussion

This chapter mainly presents the results in three parts. The first part is the results of the SAR image threshold segmentation using five different methods (see Figures 7 and 8). The training and test results of the baseline UNet and five FMNet models are introduced in the second part (see Figures 9 and 10). The test results of the oil spill image segmentation are shown in Figure 11.

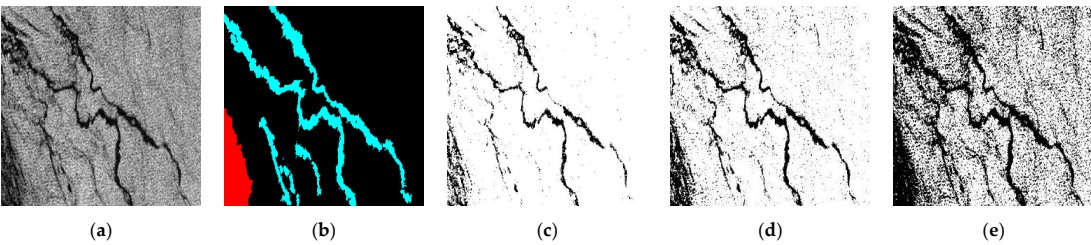

| (a) | (b) | (c) | (d) | (e) |

**Figure 7.** (**a**) The original image, (**b**) the label of the original image, (**c**) the segmentation result using a threshold of 40, (**d**) the segmentation result using a threshold of 75, and (**e**) the segmentation result using a threshold 125. The higher the threshold is set 75, the more speckle noise is retained. Setting the threshold to 75 not only removes most of the spots, but also has little impact on the oil spill area and oil spill like area (dark area).

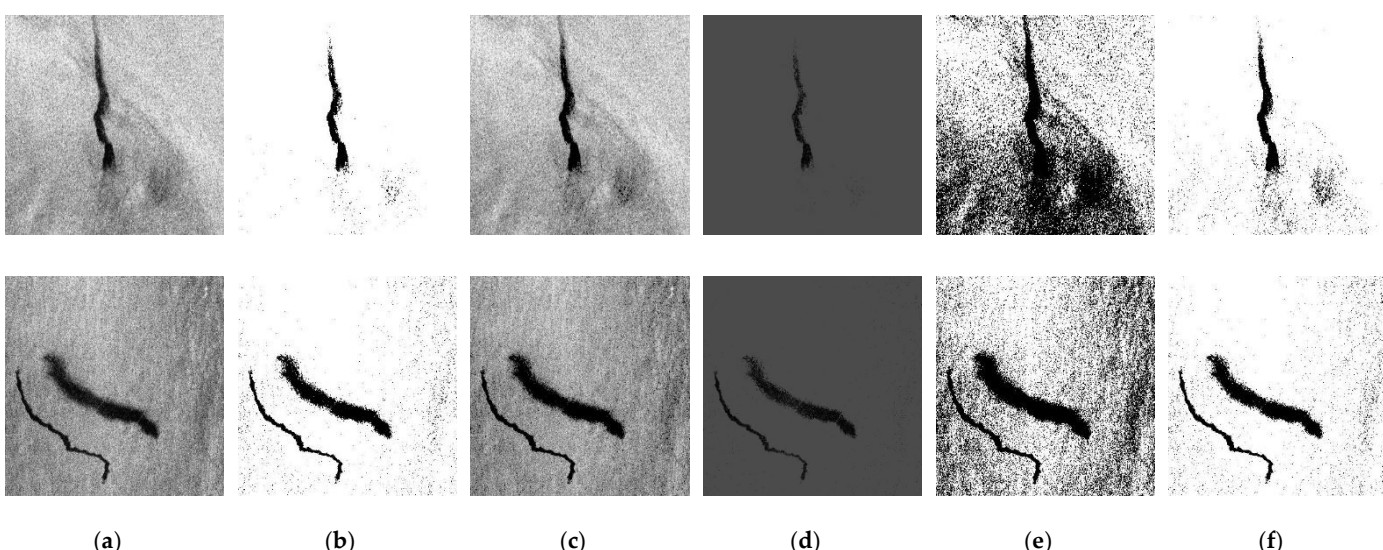

(**a**)          (**b**)          (**c**)          (**d**)          (**e**)          (**f**)

**Figure 8.** (**a**) Original image, (**b**) Binary segmentation, (**c**) ToZero segmentation, (**d**) Truncate segmentation, (**e**) OSTU segmentation, and (**f**) Triangle segmentation. The four processing methods of (**b,d–f**) are mainly for bright areas. (**c**) is mainly to strengthen the characteristics of dark areas.

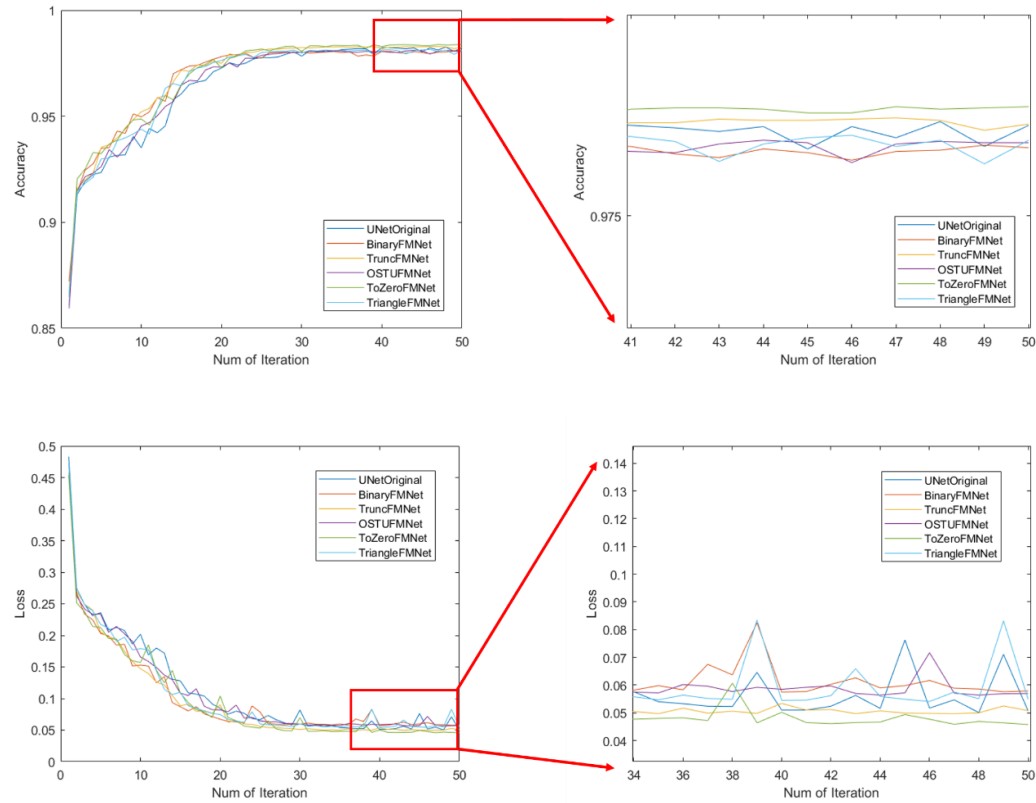

**Figure 9.** Curve of accuracy and error during training iteration.

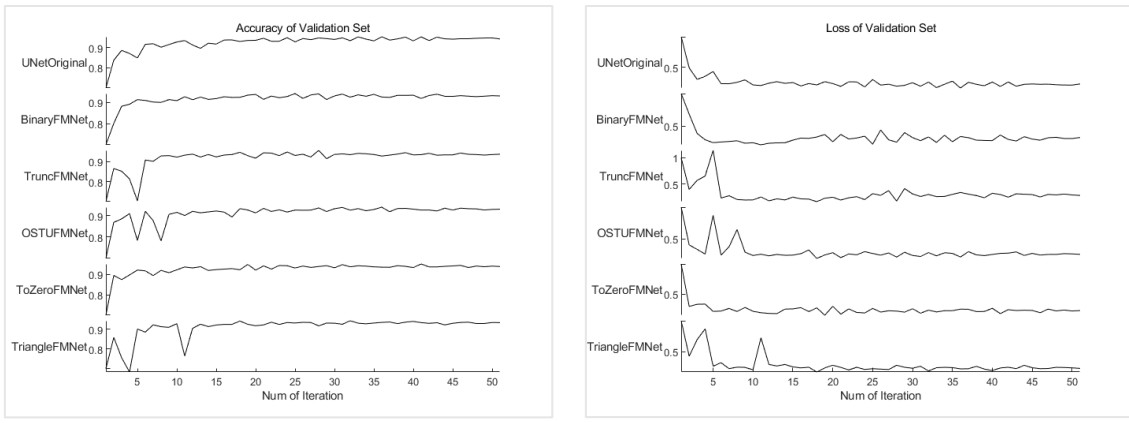

**Figure 10.** Curve of accuracy and error during validation iteration.

**Figure 11.** *Cont.*

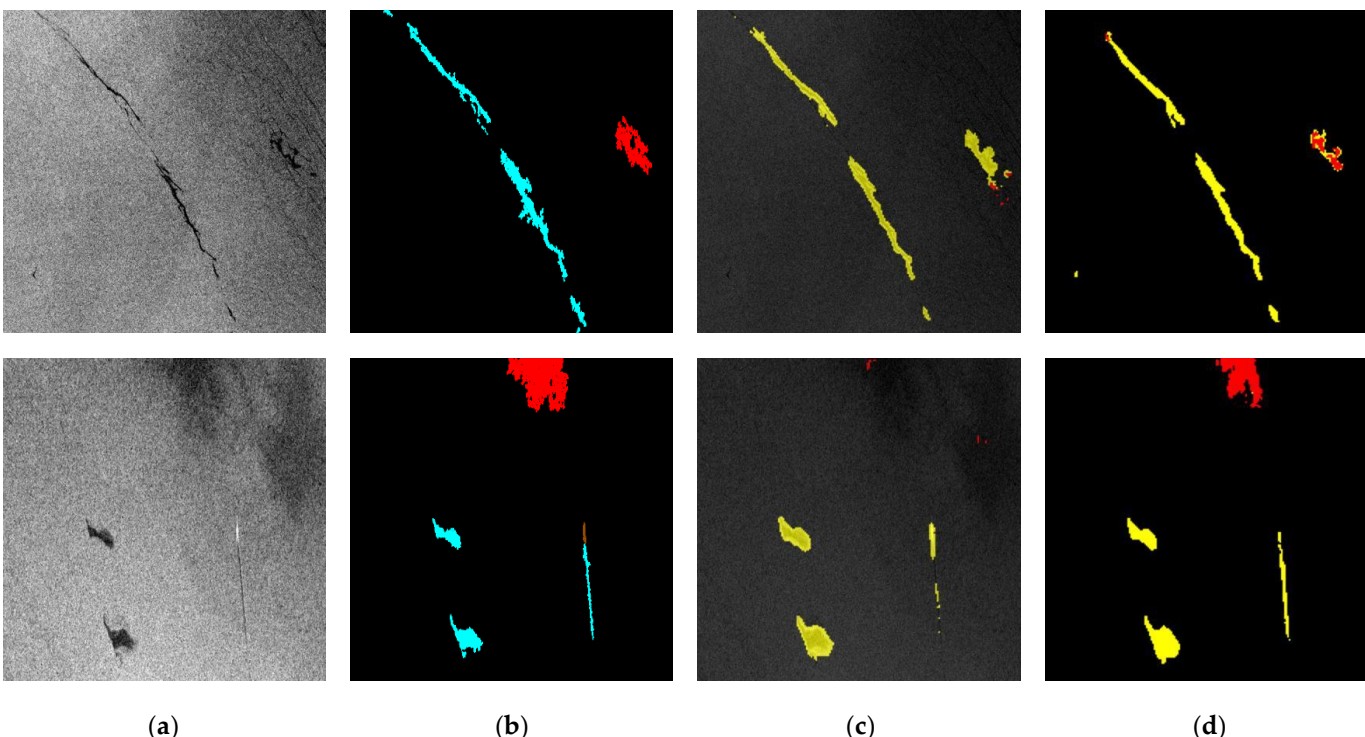

|     |     |     |     |
| :-: | :-: | :-: | :-: |
| (**a**) | (**b**) | (**c**) | (**d**) |

**Figure 11.** (**a**) SAR images, (**b**) Labels, (**c**) Segmentation results by baseline method, (**d**) Segmentation results of the proposed method. There are more errors in the recognition effect (**c**) for UNet compared with (**d**) for FMnet.

### 3.1. SAR Image Threshold Segmentation

Among these five different methods, Triangle and Otsu are adaptive threshold segmentation methods, both of which use their own algorithms to automatically select the appropriate threshold, whereas Binary, Truncate, and ToZero are manual threshold adjustment methods, and thus the size of the threshold needs to be determined first. In the experiment, 40, 75, and 125 thresholds were used to determine the optimal threshold. In terms of the effect, the three thresholds can obtain the texture of the oil spill area or area similar two an oil spill; however, when the threshold is set to 40, as shown in Figure 7c, the algorithm filters out more features with noise, and only retains the most central part of the dark area, which loses some of the features in the edge between the dark and bright regions. When the threshold is set to 125, as shown in Figure 7e, too much noise is retained, which cannot meet the design requirements of this experiment. When the threshold value is 75, as shown in Figure 7d, it not only keeps the trunk of the oil spill and oil-spill-like areas, but it also filters out most of the noise.

After the 75 thresholds were determined, 5 threshold segmentation methods were used to segment the SAR image, the results of which are shown in Figure 8b–d), which represent the results obtained by three custom threshold segmentation methods: binary, ToZero, and Truncat. In addition, Figure 8e,f shows the results obtained by the two adaptive threshold segmentation methods, OTSU and triangle. It can be seen from the results that the segmentation effects of binary and triangle are similar and are the most different from the original image in Figure 8a. Because the pixel value of the oil spill and the oil-spill-like areas is low, and most areas of the image are brighter, the custom threshold is set lower, and the pass rate of the pixel value in the binary algorithm is high. In addition, the overflow of the dark region of the oil and oil- spill-like areas are retained. The triangle algorithm uses the histogram feature to automatically find the best threshold of an image, which varies for images with different levels of brightness. Two algorithms, Binary and Triangle, filter the pixel values of the oil spill and oil-spill-like areas, highlighting the characteristics of bright areas and reducing the speckle noise of the image. By contrast, the ToZero algorithm mainly strengthens the characteristics of the dark area, uniformly changing the value of

the dark area to zero, and does not process the other areas; thus, the results are extremely similar to the source image in Figure 8a. Unlike other strategies, the Truncate algorithm uniformly changes the bright areas to greater than the threshold to be equal to the threshold, while keeping the other dark areas less than the threshold unchanged. As a result of this change, the pixel value between the dark and bright areas is reduced, and may be in the wrong direction, misleading the model. The OSTU algorithm is an adaptive threshold segmentation method. It can be seen from Figure 8e that, compared to the original image, the areas with slightly dark features are significantly enhanced, and only the areas with obvious bright features are retained, indicating that a characteristic of the threshold method is to highlight the dark areas and weaken the light areas.

### 3.2. Oil Spill Dataset Model Training

Using 90% of the images in the training dataset to train the model, the remaining 10% of the dataset was used as the validation set of the training model to verify the robustness of the model. The data are then input into the compiled model for the training process, and each iteration process will verify the prediction accuracy of the model. The loss value and accuracy rate of the training and verification processes are shown in Figure 9. The figure shows the performance of the UNet prototype and five FMNet network structures on the training set. From the perspective of the changing state of accuracy, the overall trend is the same. From the very beginning, the accuracy rate was approximately 0.86, and the accuracy rate quickly reached approximately 91% during the early stage of training, and the fluctuation then rose to approximately 98% in the later stage. The figure on the right shows the change in accuracy of the last 10 models. From Table 1, it can be seen that the final training accuracy of the UNetOriginal model is 98.16%, the loss value is 0.051, the verification accuracy rate is 93.2%, and the loss value is 0.235, which are considered baselines. The accuracy of the three models BinaryFMNet, OSTUFMNet, and TriangleFMNet are 98.02%, 98.06%, and 98.08%, respectively, which are almost the same as those of the original model; however, these models have negative characteristics in comparison to the original model, i.e., the volatility changes in the later period are severe. The ToZeroFMNet model not only achieves the best stability, but also the highest accuracy, reaching 98.40%. In addition, TruncFMNet also achieves a good stability, but its accuracy rate of 98.26% ranks second. The change trend of the loss value of each model is almost the same as that of the accuracy, although the numerical value is the opposite. From Figure 8, it can also be seen that the loss value has the best stability in the later ToZeroFMNet model, and the lowest loss value is 0.047. Several other models are volatile at this stage. Larger volatility indicates that the model experiences a serious overfitting.

**Table 1.** Comparison of training and verification results between six models. N.B. the bold text indicates the best results.

| Model Name | Phase | Accuracy (%) | Difference (%) | Loss |
|---|---|---|---|---|
| UNetOriginal | Training<br>Validation | 98.16<br>93.27 | 4.89 | 0.051<br>0.235 |
| BinaryFMNet | Training<br>Validation | 98.02<br>93.10 | 4.95 | 0.058<br>0.321 |
| TruncFMNet | Training<br>Validation | 98.26<br>93.64 | 4.62 | 0.051<br>0.284 |
| OSTUFMNet | Training<br>Validation | 98.06<br>93.15 | 4.93 | 0.057<br>0.255 |
| ToZeroFMNet | Training<br>Validation | **98.40**<br>**94.04** | **4.36** | **0.045**<br>**0.230** |
| TriangleFMNet | Training<br>Validation | 98.08<br>93.21 | 4.87 | 0.054<br>0.231 |

Figure 10 shows the change process of the accuracy and loss value of the training model on the validation set. Through a comparison, three models, UNetOriginal, BinaryFMNet, and ToZeroFMNet, were screened out. They performed better than the other three models and showed different stabilities during the process. In addition, TruncFMNet, OSTUFMNet, and TriangleFMNet experience large jitter at the beginning of the training, regardless of the accuracy or loss, indicating that the model has a staged instability. The accuracy change curve of the BinaryFMNet model is the best during the early stage, but there is a period of severe fluctuations in the middle of the progression, and at this point, ToZeroFMNet shows more lasting stability. In addition, during the early stage, it has an accuracy similar to that of the BinaryFMNet Trend. From the numerical results, the accuracy rate of the ToZeroFMNet model on the validation set was 94.04, which was the highest, and the loss value was 0.230, which was also the lowest. As the most important aspect, the differences in the accuracy and loss value of the model between the training and validation sets are the smallest, indicating that the model alleviates the overfitting phenomenon of the original approach because the smaller the difference, the lower the overfitting effect.

Among the FMNet models built on the modified of end-to-end network, ToZeroFMNet achieved the best results and alleviated the overfitting problem, which is an improvement over the original model. At the same time, the performances of the other threshold segmentation models are almost the same as those of the original model. Based on the principle of the ToZero threshold segmentation model, the ToZero model is used to strengthen the processing of the dark area of the SAR image, that is, to highlight the characteristics of the oil spill and oil-spill-like areas, whereas the other areas are not processed. The model obtained through the segmentation method Truncate, whose treatment is almost the same as that of ToZero, and UNet is also more accurate than the original model. The difference here is that the truncated threshold segmentation model was used to unify the non-dark areas to the threshold. Binary uses a threshold to divide the dark and non-dark areas more extremely. The dark areas were all set to zero, and the non-dark areas were set to 255. As the final result, the model fused with UNet had the worst effect. This may be because the threshold segmentation destroys the original numerical correlation in the images. This is only a binary distinction, and the feature parameters that can be provided to the network model are insufficient. From the above results, it can be concluded that when the threshold segmentation strengthens the characteristics of a certain category, it is not able to distinguish the categories at the same time. In this way, it can provide more parameter characteristics to the model, and better results can be obtained.

### 3.3. Test Results of the Model

After analyzing the effects of the fusion model in the training and verification stages, the model was evaluated on the test set, and 110 SAR images in the test set were segmented using the network structure obtained through training. Figure 11 shows a partial display of the results. The segmentation results obtained by the FMNet algorithm combining ToZero and UNet are more detailed than the baseline model, and the recognition results of some areas are even more accurate than the shape in the label. The original model had a relatively large segmentation error for the oil-spill-like areas. The two segmentation results in the example failed to identify these areas.

In terms of visual effects, the fusion model achieved a better performance. It also showed better results than the original model in terms of numerical statistics. The model performance is measured in terms of intersection-over-union (*IoU*), which is described as

$$IOU = \frac{prediction \cap ground\ truth}{prediction \cup ground\ truth} = \frac{TP}{FP + TP + FN} \qquad (7)$$

where *TP*, *FP*, and *FN* denote the number of true positive, false positive, and false negative samples, respectively. The *IoU* was measured for every class (five in total) of the dataset. Because the Sea Surface category occupies 88.32% of the number of pixels in the dataset, this proportion will lead to unconvincing conclusions when calculating the mean intersection

over the union. This study uses the recognition results of the other four categories to calculate the weighted *MIoU*:

$$MIoU = \frac{1}{K} \sum_{i=1}^{K} P_i * Iou_i \qquad (8)$$

where $K$ represents the number of classes, $P_i$ is the weight of $i$th class.

Another measurement is Recall defined as Equation (9). This parameter represents the number of pixels that are correctly predicted, which accounts for the proportion of the total number of pixels of the category in the sample. And *IoU* represents the intersection of the number of correct predictions in the number of samples of this type and the number of predictions of this type. The large values of these two parameters indicate that in the prediction results, there are more correct prediction results and fewer incorrect prediction results.

$$Recacll = \frac{TP}{TP + FN} \qquad (9)$$

As can be seen in Table 2, OSTUFMNet and ToZeroFMNet rank first and second with almost the same recognition accuracy on the sea surface, at nearly 1% higher than that of BaselineUnet. For the recognition of the two categories of oil spill and oil-spill-like areas, the ToZeroFMNet model ranked first at 3.46% and 1.93% higher than BaselineUNet, respectively. In terms of ship recognition, the BaselineUNet model still has the highest accuracy rate, whereas in terms of land recognition, BinaryFMNet ranks first at 2.29% higher than the BaselineUNet model, and the ToZeroFMNet model ranks second at 1.42% higher than the original model. The ToZeroFMNet model achieved a weighted MIoU accuracy rate of 61.90, which was the highest among the six models. In the performance of the six models in the recall parameters, ToZero performed best in categories sea surface, oil spill and like-oil spill, reaching 98.29%, 56.33% and 44.61% respectively, 1.42%, 2.56% and 6.85% higher than the baseline. In the performance of the Ship category, the original model still maintains the best performance. But in the recognition of the land category, Binary got a better result.

**Table 2.** Comparison between Baseline_UNet and five FMNets in terms of intersection-over-union or IoU (%). N.B. the bold text indicates the best results.

| Classes | Evaluation Index | Baseline Unet | Binary FMNet | OSTUFM Net | ToZero FMNet | Triangle FMNet | TruncFM Net |
|---|---|---|---|---|---|---|---|
| Sea Surface | IoU | 93.74 | 93.84 | **94.53** | **94.53** | 91.28 | 92.78 |
| | Recall | 96.87 | 97.27 | 98.03 | **98.29** | 94.30 | 98.27 |
| Oil Spill | IoU | 46.49 | 49.56 | 48.11 | **49.95** | 40.98 | 41.52 |
| | Recall | 53.77 | 56.09 | 56.31 | **56.33** | 41.80 | 43.51 |
| Like-oil spill | IoU | 39.47 | 37.73 | 37.95 | **41.40** | 33.75 | 35.92 |
| | Recall | 37.76 | 38.93 | 43.03 | **44.61** | 29.25 | 42.16 |
| Ship | IoU | **33.44** | 23.07 | 15.35 | 25.44 | 21.57 | 14.47 |
| | Recall | **29.63** | 23.23 | 17.00 | 21.38 | 14.31 | 2.69 |
| Land | IoU | 85.69 | **87.98** | 85.35 | 87.11 | 85.24 | 28.40 |
| | Recall | 91.27 | **95.70** | 92.79 | 93.92 | 92.09 | 34.73 |
| | MIoU | 60.08 | 60.48 | 59.29 | **61.90** | 56.64 | 30.09 |

The proposed model not only improves the recognition accuracy, it also achieves a better performance in alleviating the overfitting problem of the model. Therefore, the smaller the difference is between the training set and the test set, the less obvious the overfitting problem of the model. The training and verification results of the new algorithm on the training set were 0.4% higher than those of the original model, whereas the recognition accuracy on the test set was increased by 3.5%. With a similar performance on

the training set, there was a relatively different performance on the test set. This difference is an external manifestation of the model overfitting phenomenon and shows that the new model for the training set and the original model have almost the same number of pixel values recognized correctly, whereas for the test set, the new model has more pixel values recognized correctly than the original model.

## 4. Conclusions

Accurate and timely identification of oil spills is of significance for the treatment of marine environmental pollution. The most effective way to monitor marine oil spills is to use SAR images, because the difference in backscattering capabilities between the smooth surface of an oil spill area and the surface of the sea causes their performance on the SAR image to differ. However, oil spill areas and oil-spill-like areas have similar image characteristics, which is also an important factor that affects the accuracy of oil spill area identification. Owing to its powerful image feature mining capabilities, deep learning has made considerable progress in helping to ocean oil spills. The idea of UNET model is to provide decision-making for the segmentation of the up sampling process by using different levels of features in the convolution process. Through the experiment and analysis of image segmentation algorithm the UNet, the speckle noise in SAR image will affect the effect of image feature extraction in the process of feature extraction. Based on this, this paper proposes a new feature fusion network structure, which improves the accuracy of the original model and alleviates the over fitting problem of the original model.

Deep learning has powerful image feature mining capabilities, particularly when convolution operations are introduced into image processing. It is precisely owing to this powerful learning ability that some noise in the image will provide some out-of-order information, and this feature may lead to overfitting problems in the model. There is much speckle noise in SAR images, and such noises may provide a lot of invalid information or even interference information in the model. Based on this, five traditional threshold segmentation algorithms are used to analyze the pixel value differences of various categories of SAR images, and the pixel value of 75 is used as the threshold to remove the speckle noise in the SAR image, and at the same time it highlights the difference between the bright and dark areas and provides the model with global and concise features. The network model obtained better results after a fusion of the features, indicating that this feature fusion model has a theoretical foundation and practical value.

The use of deep learning for marine oil spill monitoring has been the main method for a long time. Although the recognition accuracy of the network structure proposed in this paper has been further improved compared to the original model, the recognition accuracy still needs further improvement. This requires a continuous exploration of the potential of the deep learning network model. The characteristics of different network structures need to be analyzed and leverage the advantages of different algorithms to obtain multiple features of data to provide better features for model decision-making to provide better results. In addition, this can adjust the structure of the original model based on the data characteristics, allowing the new model to maximize the advantages of the data themselves while avoiding their defects. Using different algorithms to obtain different features of data is an important direction for the applicationand development of deep learning to improve marine oil spill identification using SAR images.

**Author Contributions:** Methodology, Y.F., X.R.; software execution, Y.F., G.Z.; data collection, Y.F., T.Y., X.X.; writing, Y.F., T.Y.; system and method conceptualization, review and editing, X.R., S.P. All authors have read and agreed to the published version of the manuscript.

**Funding:** This research was funded by the National Key Research and Development Program of China (Grant No. 2019YFC1804304), the National Natural Science Foundation of China (Grant No. 41771478) and the Fundamental Research Funds for the Central Universities (Grant No. 2019B02514).

**Institutional Review Board Statement:** Not applicable.

**Informed Consent Statement:** Not applicable.

**Data Availability Statement:** The Oil Spill Detection Dataset we processed is from the European Space Agency (ESA) and is available online at https://m4d.iti.gr/oil-spill-detection-dataset/ (accessed on 1 May 2020).

**Acknowledgments:** We appreciate for the high-quality work done by the European Space Agency (ESA). They provided a general dataset for the field of oil spill detection, which contains a large number of SAR images of oil spill incidents (see the Data availability statement).

**Conflicts of Interest:** The authors declare no conflict of interest.

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
