# Peer review of "Feature Merged Network for Oil Spill Detection Using SAR Images"

_remotesensing, doi:10.3390/rs13163174_

Round 1

Reviewer 1 Report

An effort is done for extraction of oil spills from SAR images. Through several models achieved is increased accuracy in detection of oil spills for a few percentages. This is a certain achievement.  

Just a suggestion for TITLE:

Feature Merged Model For Oil Spill Detection Within SAR Images

MATERIAL

Satellite data from the ESA database are taken for testing. It can be understood that all the images used for testing are not listed, but the link to the dataset should be provided. It is also necessary to mention what are the dates and times of example SAR images analyzed and presented at the figures 6, 7 and 10, together with the geographical frame.

METHODS

Historical methods are well addressed, however the chapter Methods should be divided in two parts to differentiate strictly between what the others have applied from what the authors have done.

Line 448, formula 9, it is written Reacll instead of Recall.

RESULTS AND DISCUSSION

In figure legends more information should be given, particularly for figures 5, 8, 9 and 10.

The authors did not refer to any specific characteristics of pixels related to the models proposed. What does it mean physically, playing with the range of pixel values within images? Is it possible to distinguish what kind of physical characteristics these models recognize best? If these models are not only mathematical constructs, they should be connected with reflective properties of the spills.

CONCLUSIONS

This part is too general and too short. It should be rewritten. It would be best to write it as a  Summary, and should contain all what is done in the paper, concisely. Also, some ideas for future work on this topic can be given.

What is the achievement if look-alikes still cannot be distinguished from the true oil spills?

REFERENCES :

References should be carefully checked and corrected. In instructions to authors, it is not written that more than three authors should be cited as „et. al.“ , while here often one author with et al. is cited. I believe all authors should be mentioned under References.

The reference Fabio et al. 2000 at page 2, line 68 is not listed under references. Besides, this is cited in a wrong way. The author is Fabio Del Frate (Fabio is the first name), so it should be cited as :

Del Frate, F., A. Petrocchi, J. Lichtenegger and G. Calabresi, "Neural networks for oil spill detection using ERS-SAR data," in IEEE Transactions on Geoscience and Remote Sensing, vol. 38, no. 5, pp. 2282-2287, Sept. 2000, doi: 10.1109/36.868885.

In the text at page 2 line 71, mentioned is Fiscella (7) which contradicts with the reference (7) Solberg, Brekke, Husoy. 

To be corrected: Reference (27) in the text is related to ESA, but under References it is Krestenitis.

The references, most of which are from recent years should contain its' DOI, if available, although this is not requested under instructions. But some journals are sometimes behind latest developments.

Reviewer 2 Report

The paper proposed a feature merged network (FMNet) to distinguish oil spill from oil-spill-like based on SAR images. The paper is well organized and the results are also clear but there are still some problems needed to be modified. I think the paper could be accepted after serveral minor modifications.  The suggestions are as follows,

Major modifications:

1, In the paper, figure 4 is missing and please add the figure.

2, The main problems of this paper is to distinguish oil spill from oil-spill-like features or images. The main results of this paper shows in table 3 using two parameters of IOU and MIoU. Actually we are more concerned about real oil detection rates among all detect targets, false-alarm-rate of oil spill among the detected targets and how many oil spill was not detected by the proposed methods because the other two cases such as land and ship are not real concerned generally. I hope such results could be added in the paper or in the discussion. 

other suggestions:

1, Line 20, the 'UNet' should give full name in the first time appeared in the paper. 

    Line 21, 'alleviate' should be 'alleviating'. 

    Line 26, 'combines' should be 'combined'.

    Line 28-29, 'the accurary increases 1.82%' should give the final accuracy and tell us the result is  compared with which method. 

    Line 62, 'the areas with similar appearance with oil spill'。 Here it is not clear for what kinds of phenomenon could give similar features as oil spill. 

    Line 76, FCM and SC should give full name for the first time appeared in the paper. Similar problems need to be corrected in the whole paper.

   Line 135-136, the paper uses average filter to remove the salt and pepper noise. Actually SAR image owns speckle noise which belongs to coherent noise.  Lee filter is much bettter for SAR speckle noise. 

   Line 164-165 describes the proposed network but there is no framework of this proposed network. It is suggested to give a schematic diagram of this proposed netwrok. Figure 5 in page 7 shows a schematic diagram but only for image segmentation. Acctually the paper also uses FMnet to recognize oil-spill finally. 

   Line 207, Figure 4 is not included in the paper.

   Line 222, all parameters should be explained below the equations. 

   Figure 5 (Line 261) and Table 1 (line 294) should give a full network for image segmentation and classification using deep learning algorithm. 

   Line 282, 'Keras framework' needed add a reference. 

   Line 315, 'the threshold for 40, 75 and 125', Does all sar image have been processed and let the data varied from 0 to 255?  How to determine such threshold value?

   Line 440,  how to define the parameter of Pi in equation (8) ?

Reviewer 3 Report

see the attached for changes
